# Defects in immune response to *Toxoplasma gondii* are associated with enhanced HIV-1-related neurocognitive impairment in co-infected patients

**Edwin Eliel Escobar-Guevara**[1,2,3]*, **María Esther de Quesada-Martínez**[4], **Yhajaira Beatriz Roldán-Dávila**[5,6], **Belkisyolé Alarcón de Noya**[7], **Miguel Antonio Alfonzo-Díaz**[1,8,9]

1 Laboratory of Cellular Immunophysiology, José Maria Vargas School of Medicine, Central University of Venezuela, Caracas, Venezuela, 2 Department of Immunology, José Maria Vargas School of Medicine, Central University of Venezuela, Caracas, Venezuela, 3 Laboratory of Physiopathology, Venezuelan Institute for Scientific Research, Caracas, Venezuela, 4 Department of Physiopathology, José Maria Vargas School of Medicine, Central University of Venezuela, Caracas, Venezuela, 5 Service of Infectology, José Ignacio Baldó Hospital, Caracas, Venezuela, 6 Department of Microbiology, José Maria Vargas School of Medicine, Central University of Venezuela, Caracas, Venezuela, 7 Tropical Medicine Institute, Section of Immunology, Central University of Venezuela, Caracas, Venezuela, 8 Department of Physiology, José Maria Vargas School of Medicine, Central University of Venezuela, Caracas, Venezuela, 9 Academic Department, Salvador Allende Latin-American School of Medicine, San Antonio de Los Altos, Miranda State, Venezuela

* edscobar@gmail.com

## Abstract

Human immunodeficiency virus-1 (HIV-1) and *Toxoplasma gondii* can invade the central nervous system and affect its functionality. Advanced HIV-1 infection has been associated with defects in immune response to *T. gondii*, leading to reactivation of latent infections and development of toxoplasmic encephalitis. This study evaluates relationship between changes in immune response to *T. gondii* and neurocognitive impairment in HIV-1/*T. gondii* co-infected patients, across different stages of HIV-1 infection. The study assessed the immune response to *T. gondii* by measuring cytokine production in response to parasite antigens, and also neurocognitive functions by performing auditory and visual P300 cognitive evoked potentials, short term memory (Sternberg) and executive function tasks (Wisconsin Card Sorting Test-WCST) in 4 groups of individuals: HIV-1/*T. gondii* co-infected (P2), HIV-1-infected/*T. gondii*-non-infected (P1), HIV-1-non-infected/*T. gondii*-infected (C2) and HIV-1-non-infected/*T. gondii*-non-infected (C1). Patients (P1 and P2) were grouped in early/asymptomatic (P1A and P2A) or late/symptomatic (P1B/C and P2B/C) according to peripheral blood CD4+ T lymphocyte counts (>350 or <350/μL, respectively). Groups were compared using *T-student* or *U-Mann-Whitney* tests as appropriate, $p<0.05$ was considered as significantly. For P300 waves, HIV-1-infected patients (P1) had significantly longer latencies and significantly smaller amplitudes than uninfected controls, but HIV-1/*T. gondii* co-infected patients (P2) had significantly longer latencies and smaller amplitude than P1. P1 patients had significantly poorer results than uninfected controls in Sternberg and WCST, but P2 had significantly worse results than P1. HIV-1 infection was associated with significantly lower production of IL-2, TNF-α and IFN-γ in response to *T. gondii* from early/

**Data Availability Statement:** All relevant data are within the paper and its Supporting Information files.

**Funding:** This work was supported by grant awarded to MAAD from FONACIT (G-2005000823) and by grant awarded to EEEG from Venezuelan Institute for Scientific Research (IVIC-2009000441). EEEG received doctoral scholarships from FONACIT and Venezuelan Institute for Scientific Research. The funders had no role in study design, data collection and analysis, decision to publish, or preparation of the manuscript.

**Competing interests:** The authors have declared that no competing interests exist.

asymptomatic stages, when comparing P2 patients to C2 controls. These findings may indicate impairment in anti-parasitic response in co-infected patients, facilitating early limited reactivation of the parasitic latent infection, therefore creating cumulative damage in the brain and affecting neurocognitive functions from asymptomatic stages of HIV-1 infection, as suggested by defects in co-infected patients in this study.

## Introduction

*Toxoplasma gondii* is an obligate intracellular protozoan parasite, causing one of the more common human parasitic infections [1], with an average global seroprevalence rate of 25.7% [2]. Infection with *T. gondii* is usually chronic and latent in immunocompetent hosts, the parasite persisting as tissue cysts, especially in the central nervous system (CNS) [3,4]. During the acute phase of infection, tachyzoites (the fast-replicating stage of the parasite) [5] quickly proliferate within a variety of nucleated cells, including neurons, astrocytes and other glial cells in the brain [6–8]; then, after activation of immune response, bradyzoites (the slow-replicating stage) [5] form cysts within infected cells, establishing the chronic infection. Production of Interferon-gamma (IFN-γ) by cells of innate [9,10] and adaptive [11,12] immunity, is a hallmark of the protective response [13], leading to activation of macrophages, production of tumor necrosis factor-alpha (TNF-α) and expression of inducible nitric oxide synthase (iNOS) [14,15], creating an environment that induces conversion to bradyzoite stage [16]. Production of IFN-γ is also essential to prevent reactivation of the chronic infection [17,18], and, although several cell populations participate in this production, the role of T lymphocytes is critical in host resistance [11,12,19,20]. In this protective immune response, interleukin-2 (IL-2) mediates expansion of T and *natural killer* lymphocytes [21]; while modulation by IL-10 is required to prevent damage of host´s tissues [22,23]. It is noteworthy that immunosuppression, as that associated with advanced human immunodeficiency virus-1 (HIV-1) infection [24,25], may result in reactivation of latent infection, causing disruption of cysts, proliferation of tachyzoites, inflammation and destruction of host's tissues. The reactivation of chronic infection in the CNS usually presents as toxoplasmic encephalitis, a major and life-threatening opportunistic disease in patients with acquired immunodeficiency syndrome (AIDS) [26]. On the other hand, HIV-1 infection can induce several neurological syndromes, including peripheral neuropathies, vacuolar myelopathy and HIV-associated neurocognitive disorders (HAND) [27–31]. Therefore, co-infection with HIV-1 and *T. gondii* poses a greater threat to neurocognitive function.

As stated before, advanced HIV-1 infection is associated with toxoplasmic encephalitis, but, as immune dysregulation is progressive [32–34], we were interested in studying how *T. gondii*-specific response changes in different stages of the viral infection, and whether this change could be associated to modifications in neurocognitive functions. Immune response was assessed by studying cytokine production in response to *T. gondii* antigens, and neurocognitive functions by evaluating auditory and visual P300 cognitive evoked potentials [35], Sternberg's speed of memory scanning task [36] and Wisconsin Card Sorting Test (WCST) [37]. With this approach, we were able to detect early defects in immune response to *T. gondii* as well as premature alterations in neurocognitive functions in HIV-1/*T. gondii* co-infected patients.

## Methods and materials

### Participants

69 HIV-1 seropositive and 14 HIV-1 seronegative adult individuals (S1 Table) were enrolled from the Service of Infectology at José Ignacio Baldó Hospital; the Section of Immunology at Tropical Medicine Institute and the José Maria Vargas School of Medicine, Central University of Venezuela, all in Caracas. All participants provided written informed consent and the study was approved by the Ethic Boards of the participating institutions, in accordance with the Declaration of Helsinki (1964–2013) and the Belmont Report (1979). Serological status for HIV-1 was determined by ELISA and Western Blot tests; while for *T. gondii*, by ELISA (IgG, IgM) and avidity tests. Enrolled participants had not received any anti-retroviral (cART) or anti-toxoplasmic therapies at the time of the study. Exclusionary criteria included a history of traumatic brain injury, autoimmune disease (i.e., systemic lupus erythematosus, rheumatoid arthritis), non-HIV-1 associated CNS disease (i.e., Alzheimer's, Parkinson's), current alcohol or illicit drugs abuse or dependence, and treatment with immunosuppressant drugs. A sample of peripheral blood, using EDTA as anticoagulant, was drawn from each participant to perform immunological studies (viral load of HIV-1, immunophenotyping of T lymphocytes, culture of cells and determination of cytokines). Neurophysiologic evaluation (P300, Sternberg and WCST) was completed within a week of blood testing. Participants were grouped as follows:

- **Control Group 1 (C1):** 5 asymptomatic individuals (2 men, 3 women; 38±4 years of age), seronegative for both HIV-1 and *T. gondii*.

- **Control Group 2 (C2):** 9 asymptomatic individuals (6 men, 3 women; 31±12 years of age), seronegative for HIV-1 and seropositive for *T. gondii*.

- **Patients Group 1 (P1):** 36 individuals (26 men, 10 women; 34±10 years of age), seropositive for HIV-1 and seronegative for *T. gondii*. These patients were grouped according to CD4+ T lymphocyte counts in peripheral blood, as follows:

  ○ **P1A Group (Early/Asymptomatic):** 11 individuals (6 men, 5 women; 31±8 years of age), with CD4+ T cell counts in peripheral blood greater than 350/μL.

  ○ **P1B/C Group (Late/Symptomatic):** 25 individuals (20 men, 5 women; 35±10 years of age), with CD4+ T cell counts in peripheral blood below 350/μL.

- **Patients Group 2 (P2):** 33 individuals (26 men, 7 women; 37±10 years of age), seropositive for both HIV-1 and *T. gondii*. These patients were grouped according to CD4+ T lymphocyte counts in peripheral blood, as follows:

  ○ **P2A Group (Early/Asymptomatic):** 9 individuals (6 men, 3 women; 33±11 years of age), with CD4+ T cell counts in peripheral blood greater than 350/μL.

  ○ **P2B/C Group (Late/Symptomatic):** 24 individuals (20 men, 4 women; 38±10 years of age), with CD4+ T cell counts in peripheral blood below 350/μL.

For neurophysiologic evaluations (P300, Sternberg, WCST), the control group consisted of 41 asymptomatic HIV-1 seronegative individuals (21 men, 20 women; 47±9 years of age), who share similar characteristics to those of other participants [38,39].

## Culture of peripheral blood mononuclear cells

Peripheral blood mononuclear cells (PBMC) were obtained by density gradient isolation (Histopaque 1077; Sigma-Aldrich) and cultured (1 x $10^5$ cells/well) for 72 hours, at 37˚C, 5% $CO_2$, in triplicate, in 96 well culture plates (Nunclone, Nunc), in a final volume of 200 μL/well, in complete RPMI medium (RPMI 1640-Gibco; HEPES buffer-Gibco, 10 mM; sodium pyruvate-Sigma, 1 mM; non-essential amino acids-Gibco, 1x; L-glutamine-Gibco, 2 mM; antibiotics-Gibco: penicillin, 100 U/mL; streptomycin, 100 mg/mL; fetal bovine serum-Gibco, 10%), in three different conditions:

- **Medium:** PBMC cultured in basal conditions, with no stimulation.

- **PHA:** PBMC cultured in the presence of the polyclonal activator phytohemagglutinin (PHA, Sigma, 5 μg/mL)

- **SATg:** PBMC cultured in the presence of *T. gondii* soluble tachyzoites antigen (SATg, provided by Belkisyolé Alarcón de Noya, Tropical Medicine Institute, Central University of Venezuela, Caracas; 1 μg/mL)

After 72 hours, culture supernatants were collected and stored at -80˚C, until cytokines were determined.

## Determination of cytokines

Levels of IL-2, IL-10, TNF-α and IFN-γ were determined in culture supernatants by flow cytometry, using Cytometric Bead Array Human Th1/Th2 Cytokine Kit (Becton-Dickinson), following manufacturer's instructions. Briefly, culture supernatants were mixed with several bead populations with distinct fluorescence intensities; each bead population had been coated with antibodies specific for IL-2, IL-10, TNF-α or IFN-γ, so cytokines in sample were captured by beads. Simultaneously, samples were incubated with phycoerythrin (PE)-conjugated detection antibodies, also specific for each cytokine, so sandwich complexes (capture bead + cytokine + detection antibody) were formed. After proper incubation, samples were acquired in a flow cytometer (FACScalibur, Becton Dickinson). Bead populations, representing each kind of cytokine, were resolved in the FL3 fluorescence channel, and cytokine levels were resolved in the FL2 (PE) fluorescence channel. It was possible to calculate levels of cytokines in samples using standard curves.

## P300 cognitive evoked potentials

The electroencephalography (EEG) signals were recorded using a Medicid 4 Plus, Neuronic equipment, with 20 monopolar gold-disk electrodes placed according to the 10–20 international system [40] including Oz position, with impedance lower than 5 kΩ, at a sampling rate of 200 Hz (5 ms) and band-pass from 0,5 to 30 Hz. Two additional electrodes were placed on the right and left earlobes, to be used as reference. Two electrodes were used to detect eye-movement artifact, one placed 1 cm above the left outer canthus and other placed 1 cm below the right outer canthus; subsequently, wave signals recorded within eye-movement artifact were removed from analysis. Subjects were seated comfortably when experiments were performed. Defects in auditory and visual sensorial pathways were ruled out in all participants by recording normal wave results in brainstem auditory evoked potentials, pattern reversal visual evoked potentials and electroretinogram.

Auditory and visual P300 cognitive evoked potentials were recorded using the oddball stimulation paradigm [35,41], with 130 stimuli, 25% of which were infrequent (*target stimulus*), presented at random. Participants were instructed to click the left button of the computer

mouse each time target stimulus appeared. Visual stimuli were presented on the computer screen and were represented by the figure of a horse (target/infrequent stimulus) and the figure of a bicycle (frequent/non target stimulus). Each stimulus was presented during 500 ms. Auditory stimuli were represented by a 2,000 Hz tone (target/infrequent stimulus) and a 500 Hz tone (frequent/non target stimulus), each was presented during 100 ms, at 60 dB SPL of intensity. Time between stimuli (visual as well as auditory) was 1,200 ms.

EEG data were analyzed off line. Visualization windows for analysis were set to begin 100 ms before, and to finish 800 ms after stimulus onset. Amplitude and latency were recorded. Amplitude is defined as the difference between the mean pre-stimulus baseline voltage and the largest positive-going peak of the waveform within a time window, and latency is defined as the time from stimulus onset to the point of maximum positive amplitude within a time window. Frequent and infrequent stimuli's data from each single electrode were analyzed and averaged separately.

## Evaluation of memory

Exploration of memory was achieved according to Sternberg [36]. Briefly, sets of letters (from two to six letters in white font per sequence) were presented one at a time to the participants on the computer screen, immediately followed by a single yellow letter. The participant had to indicate whether the yellow letter was in the previous set or not by pressing the key "Ins" on the computer keyboard if yes, or the key "Alt" if not. Each letter was presented during 1 second and the time between letters was 800 ms. A total of 15 sets of letters were presented and time between sets was 1,500 ms. The parameters recorded were number of right answers, number of wrong answers, absence of answers and time to answer.

## Wisconsin Card Sorting Test

A computer-based version of the Wisconsin Card Sorting Test (WCST) [37,42] was used. In brief, each WCST trial began with the display of four *key-cards* in the upper area of the computer screen, plus one *choice-card* displayed in the lower area of the screen. Subjects were asked to classify/sort the *choice-card* by selecting the appropriate *key-card* according to three classification criteria:

- Color of the figure: red, green, yellow or blue

- Shape or kind of figure: triangle, star, cross or circle

- Number of figures: one, two, three or four

After selection was made, feedback was provided by means of a "Right" or "Wrong" text box, and a new trial was presented. The classification criteria automatically changed every ten selections, with no previous notification to the participant. The total number of trials was 128. The inter-trial interval was 500 ms. The score recorded for WCST were the number of categories completed, the number of perseverative errors (all consecutive errors starting from the second trial in a set), the numbers of failures to maintain set (if subject made a wrong classification after five right selections in any set) and the total number of errors.

A computer-based cognitive diagnostic system incorporated to the Medicid 4 Plus-Neuronic equipment was used to perform WCST and memory exploration. Cognitive impairment was ruled out in all participants by previously obtaining normal scores (28–30 points) in a mini mental state examination [43].

## Statistical analysis

SigmaStat® for Windows, Version 1.0 (1992–1994 Jandel Corporation) was used to perform statistical analysis of data. Groups were compared using two-tailed *T-student* or *Mann-Whitney* tests, as appropriate. Differences were considered as statistically significant if *p* values were lower than 0.05. Pearson product moment was used to analyze correlation, *p<0.05* was considered as significantly.

## Results

### Immune response against *T. gondii*

**HIV-1/*T. gondii* co-infected patients produce no IL-2 in response to parasite's antigens.** PBMC from C2 group (HIV-1-non infected/*T. gondii*–infected individuals) produced more IL-2 than any other group (S2 and S8 Tables) when they were stimulated for 72 hours with soluble antigens of tachyzoites of *T. gondii* (SATg), with a statistically significant difference when compared to the production without stimulation (p = 0.001; Fig 1A, *SATg vs. Medium*). On the other hand, production of IL-2 by PBMC from P2 patients (HIV-1/*T. gondii* co-infected) cultured under SATg stimulation, was significantly lower than found on C2 group, both in the early and late stages (p = 0.006; p<0.001, respectively; Fig 1A, *SATg*). None of the groups incremented its production of IL-2 when PBMC were culture for 72 hours in the presence of PHA (Fig 1A, *PHA vs. Medium*).

**HIV-1-infected patients show non-specific and spontaneous production of IL-10.** Culturing PBMC from P1A group in the presence of SATg produced significantly more IL-10 than from C1 (p = 0.016; Fig 1B, *SATg*) and than from their respective cultures in medium (p = 0.029; Fig 1B, *SATg vs. Medium*). P1B/C subjects showed spontaneous production of IL-10 in non-stimulated cultures, significantly higher than P1A and C1 controls (p = 0.024 and p = 0.042, respectively; Fig 1B, *Medium*). C2 and P2B/C groups produced significantly more IL-10 under SATg stimulation than in non-stimulated cultures (p = 0.001 and p = 0.009, respectively; Fig 1B, *SATg vs. Medium*); C2 also produced significantly more IL-10 than from C1 (p = 0.040; Fig 1B, *SATg*).

**HIV-1 infection is associated with enhanced production of TNF-α under polyclonal stimulation.** Under PHA stimulation, all groups produced significantly higher levels of TNF-α (S2 and S8 Tables) than cultures in medium (p≤0.032; Fig 1C, *PHA vs. Medium*; for each p-value see S9 Table), but production of patients groups (P1 and P2) was significantly higher than production of control groups (C1 and C2, respectively) in this type of polyclonal stimulation (p≤0.029; Fig 1C, *PHA*; for each p-value see S9 Table).

**Production of TNF-α in response to parasite's antigens is significantly lower in HIV-1/*T. gondii* co-infected patients.** Under SATg stimulation C2 produced significantly more TNF-α than C1 (p = 0.004; Fig 1C, *SATg*) and than its respective culture in medium (p = 0.001; Fig 1C, *SATg vs. Medium*). Also, both groups of P2 patients produced more TNF-α under SATg stimulation than cultures in medium (significantly for P2B/C: p = 0.006; Fig 1C, *SATg vs. Medium*) but less than C2 group (significantly for P2A: p = 0.041; Fig 1C, *SATg*).

**Production of IFN-γ in response to parasite's antigens is significantly lower in HIV-1/*T. gondii* co-infected patients.** Under SATg stimulation C2, P2A and P2B/C groups produced significantly higher levels of IFN-γ than in medium (p = 0.004, p = 0.026 and p = 0.006, respectively; Fig 1D, *SATg vs. Medium*), and also higher than C1, P1A and P1B/C groups under SATg stimulation as well (significantly for C2 vs. C1: p = 0.016; and for P2B/C vs. P1B/C: p = 0.024; Fig 1D, *SATg*). However, P2A and P2B/C patients produced significantly lower

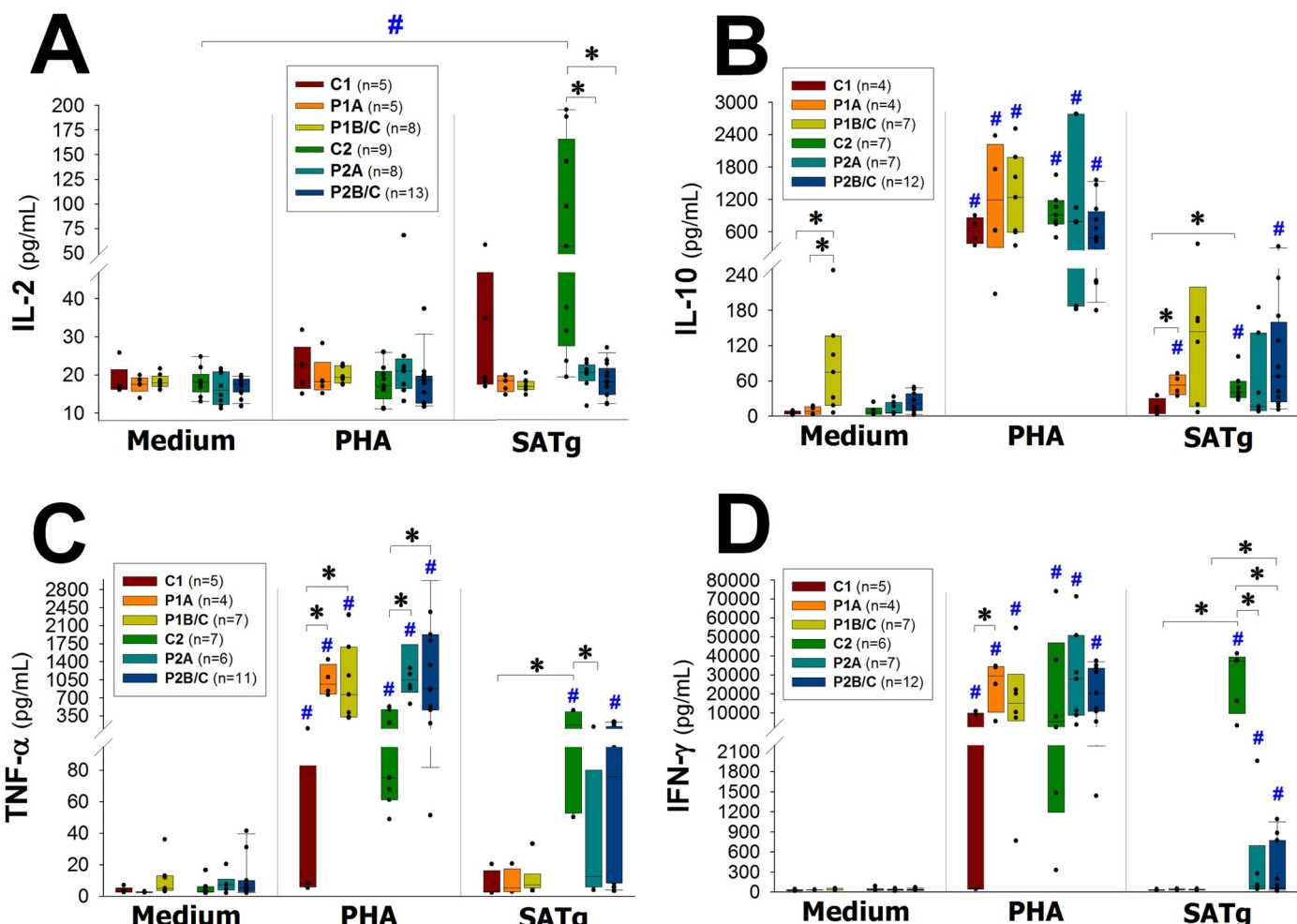

**Fig 1. Production of cytokines.** Vertical boxes with error bars represent medians, 25th and 75th percentiles of the levels (pg/mL) of IL-2 (A), IL-10 (B), TNF-α (C) and IFN-γ (D) in the supernatants of cultures of PBMC from HIV-1-non infected individuals that were seronegative (C1) or seropositive (C2) to *T. gondii*, and from HIV-1 infected (P1) or HIV-1/*T. gondii* co-infected (P2) patients. Patients (P1 and P2) were grouped in early/asymptomatic or late/symptomatic, depending on whether their peripheral blood CD4+ T lymphocytes counts were higher (P1A and P2A) or lower (P1B/C and P2B/C) than 350/μL, respectively. PBMC were cultured for 72 hours in basal/non-stimulated conditions (Medium), in the presence of the polyclonal stimulator phytohemagglutinin (PHA, 5 μg/mL) or in the presence of soluble antigens of tachyzoites of *T. gondii* (SATg, 1 μg/mL). Black dots represent every single patient; asterisks (*) represent statistically significant differences between groups, blue number signs (#) represent statistically significant differences between a group in any given condition and its respective culture in medium.

levels of IFN-γ than C2 group in cultures stimulated with SATg (p = 0.004 and p = 0.002, respectively; Fig 1D, *SATg*).

**Production of IL-2, IFN-γ and TNF-α are significantly correlated.** Upon analyzing the production of cytokines in response to *T. gondii* antigens (SATg) in individuals seropositive to the parasite (C2 and P2 groups), we found significantly positive correlations between IFN-γ and IL-2 (Pearson = 0.81; p<0.001, n = 25; Fig 2A), TNF-α and IL-2 (Pearson = 0.72; p<0.001, n = 23; Fig 2B), TNF-α and IFN-γ (Pearson = 0.80; p<0.001, n = 20; Fig 2C).

## Neurophysiologic evaluation

**HIV-1/*T. gondii* co-infected patients have longer visual P300 latencies than HIV-1-infected patients.** HIV-1-infected patients (P1 and P2) in early/asymptomatic as well as

# Cultures under SATg stimulation

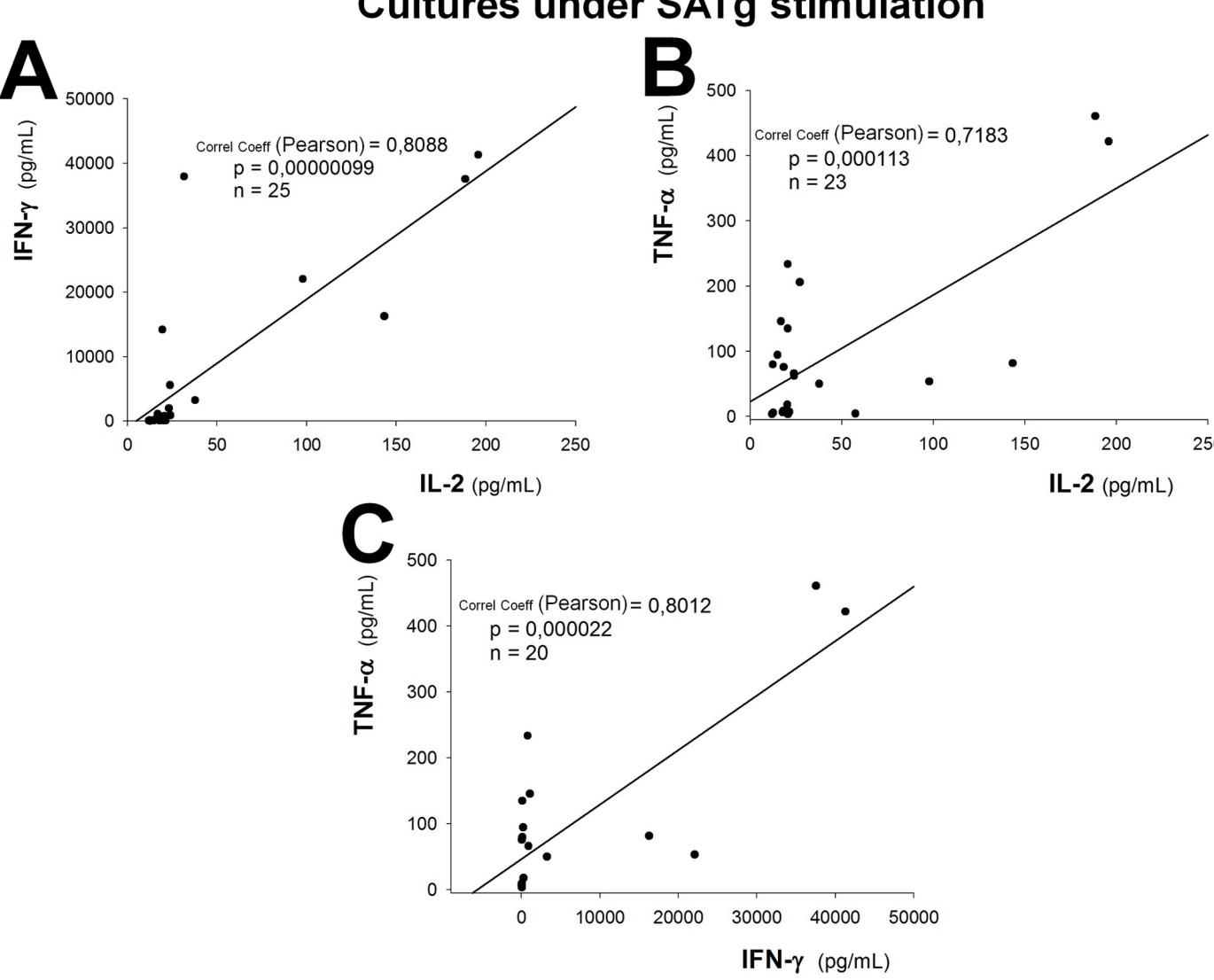

**Fig 2. Correlation between cytokines.** Correlations in the production of IFN-γ vs. IL-2 (A), TNF-α vs. IL-2 (B) and TNF-α vs. IFN-γ (C) under SATg stimulation in individuals seropositive to *T. gondii* (C2 and P2 groups) are shown.

late/symptomatic stages showed prolonged auditory and visual P300 latencies when compared to the non-infected group (S3 and S4 Tables), with statistically significant differences for visual P300 at the electrode locations Fp1, Fp2, F3, F4, C3, C4, P3, P4, O1, O2, F8, T6, Fz, Cz, Pz and Oz of the 10–20 international system [40] (p≤0.048; Fig 3A; for each p-value see S10 Table), and at all electrode locations for auditory P300 (p≤0.049; Fig 3B; for each p-value see S11 Table). On the other hand, HIV-1/*T. gondii* co-infected patients in late/symptomatic stage (P2B/C) showed significantly longer visual P300 latencies than P1B/C group at the occipital electrode locations O2 and Oz (p = 0.017 and p = 0.027, respectively; Fig 3C, *P2B/C vs. P1B/C*).

**HIV-1/*T. gondii* co-infected patients have a smaller auditory P300 amplitude than HIV-1-infected patients.** HIV-1-infected patients (P1 and P2) showed significantly smaller auditory P300 amplitudes at the frontal electrode locations F4 and Fz (S5 Table) than the non-infected group (p≤0.049; Fig 4A and 4B; for each p-value see S12 Table). HIV-1/*T. gondii* co-

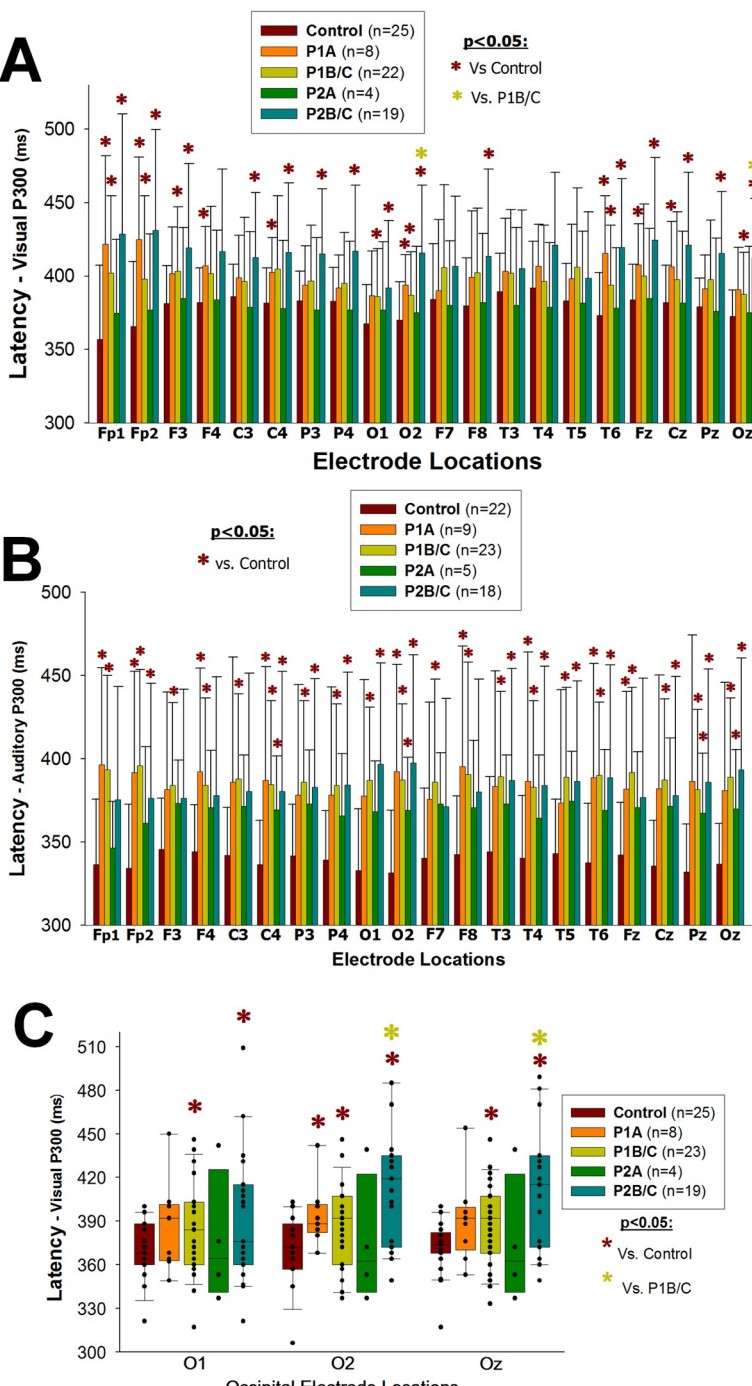

**Fig 3. P300 –latency.** Bars (A, B) represent the mean and standard deviation; vertical boxes with error bars (C) represent medians, 25th and 75th percentiles of visual (A, C) and auditory (B) P300 latencies (ms) at each electrode location, according to the 10–20 international system [40], in HIV-1-infected (P1), HIV-1/*T. gondii* co-infected (P2) and HIV-1 non-infected (Control) individuals. Patients (P1 and P2) were grouped in early/asymptomatic or late/symptomatic, depending on whether their peripheral blood CD4+ T lymphocytes counts were higher (P1A and P2A) or lower (P1B/C and P2B/C) than 350/μL, respectively. Black dots (C) represent every single patient, dark red asterisks (*) represent statistically significant differences between any given group of patients and the HIV-1 non-infected group at the same electrode location; light green asterisks (*) represent statistically significant differences between P2B/C and P1B/C at a particular electrode location.

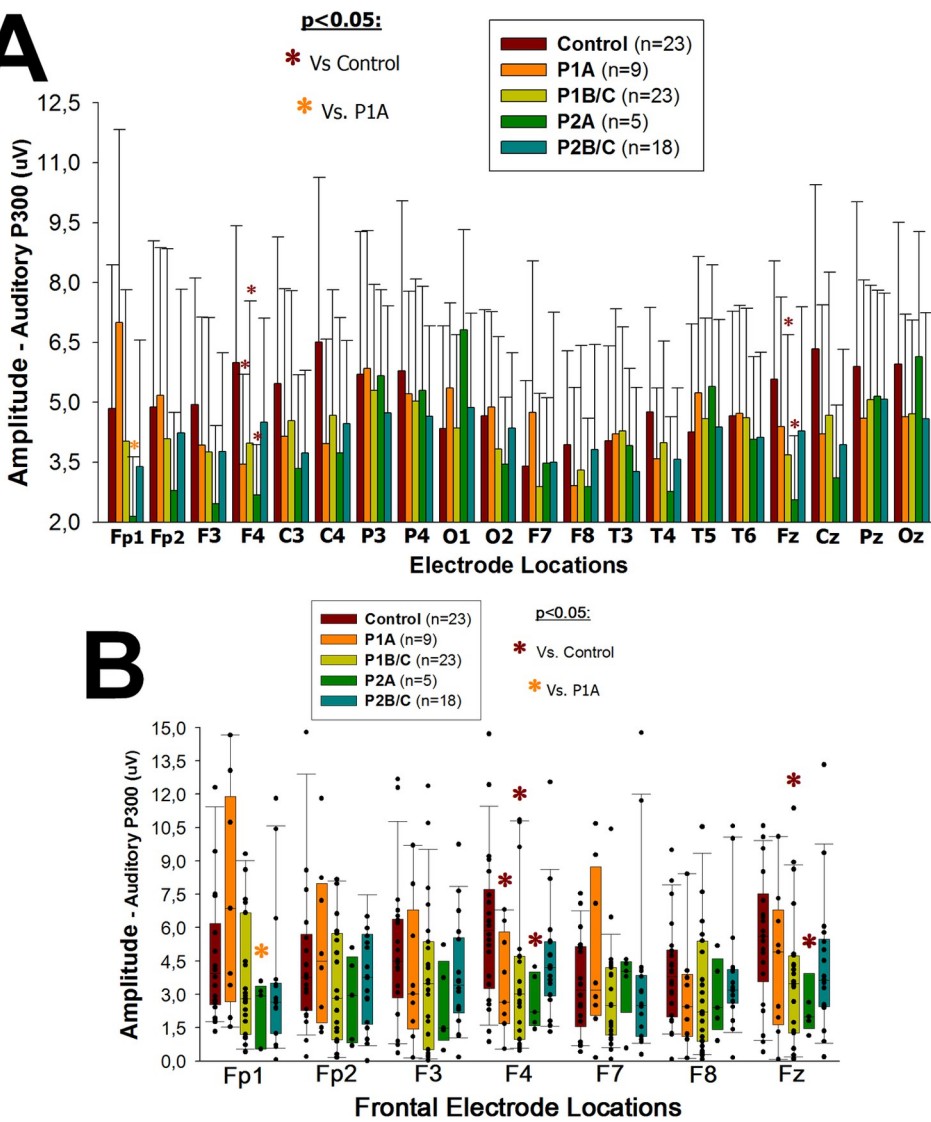

**Fig 4. P300 –amplitude.** Bars (A) represent the mean and standard deviation; vertical boxes with error bars (B) represent medians, 25th and 75th percentiles of auditory P300 amplitudes (µV) at all (A) or frontal (B) electrode locations, according to the 10–20 international system [40], in HIV-1-infected (P1), HIV-1/*T. gondii* co-infected (P2) and HIV-1 non-infected (Control) individuals. Patients (P1 and P2) were grouped in early/asymptomatic or late/symptomatic, depending on whether their peripheral blood CD4+ T lymphocytes counts were higher (P1A and P2A) or lower (P1B/C and P2B/C) than 350/µL, respectively. Black dots (B) represent every single patient, dark red asterisks (*) represent statistically significant differences between any given group of patients and the HIV-1 non-infected group at the same electrode location; orange asterisk (*) represents statistically significant difference between P2A and P1A at the electrode location Fp1.

infected patients in the early/asymptomatic stage (P2A) showed a significantly smaller auditory P300 amplitude than P1A group at the Fp1 electrode location (p = 0.046; Fig 4B, *P2A vs. P1A*).

**HIV-1/*T. gondii* co-infected patients perform worse on memory assessment than HIV-1-infected patients.** In the evaluation of memory according to Sternberg [36] (S6 Table), HIV-1-infected patients (P1 and P2) showed a significantly smaller number of right answers (p≤0.040) and a significantly greater number of absent answers (p≤0.001) than the non-

infected group (Fig 5A; for each p-value see S13 Table). HIV-1/*T. gondii* co-infected patients in the early/asymptomatic stage (P2A) showed a significantly greater number of wrong answers than P1A group (p = 0.043; Fig 5B) in tasks with the longest memory sets (sets of 6 letters).

**HIV-1/*T. gondii* co-infected patients perform worse on the Wisconsin Card Sorting Test than HIV-1-infected patients.** HIV-1-infected patients (P1 and P2) showed significantly poorer results in WCST (S7 Table) than the HIV-1 non-infected group, with significantly smaller numbers of categories completed (p≤0.008; Fig 6A; for each p-value see S14 Table) and significantly greater numbers of perseverative errors, total errors and failures to maintain set (p≤0.021, p≤0.002 and p<0.001, respectively; Fig 6B; for each p-value see S14 Table). Furthermore, HIV-1/*T. gondii* co-infected patients in the early/asymptomatic stage (P2A) showed a significantly greater number of failures to maintain set than P1A group (p = 0.030; Fig 6B).

## Discussion

Immunocompetent hosts are able to control *T. gondii* infection by inducing an early and strong cellular immune response against the parasite [44–46]. In this response, IFN-γ is the major mediator of host resistance [13], stimulating the expression of several effectors, such as IFN-inducible guanosine triphosphatases (GTPases), inducible nitric oxide synthase (iNOS) and indoleamine-2,3-dioxygenase (IDO) [47,48], that lead to vigorous cell-autonomous immune responses, with suppression of growth and/or direct killing of the parasite within infected cells. TNF-α also plays an essential role in this protective response, particularly in the brain, where it reduces entry of the parasite to the host´s cells, inhibits its proliferation and mediates its killing [14,15,49–51]. On the other hand, IL-10 modulates the response, preventing lethal immunopathology [22,23,45]. Thus, the profile of cytokines produced in response to *T. gondii* is critical for an effective yet safe control of infection [44–46]. In the experiments described in this work, individuals seropositive for *T. gondii* but seronegative for HIV-1 produced significantly higher levels of IFN-γ, TNF-α and IL-10 in response to parasite antigens than individuals seronegative for both *T. gondii* and HIV-1, and also produced a significantly higher level IL-2, an important cytokine for T cell function and regulation [21,52,53]. Therefore, it was remarkable that co-infected patients, even in early stages of HIV-1 infection, had produced significantly lower levels of IL-2, TNF-α and IFN-γ under SATg stimulation, associating HIV-1 infection with early defects in the response against *T. gondii*, which could impair the control of latent infection in the CNS, allowing limited reactivation, that is, not associated with toxoplasmic encephalitis, but capable of generating cumulative damage and functional impairment, even in the asymptomatic stage of HIV-1 infection.

Neuroelectric measures can provide direct imaging of CNS function [54]. The P300 brain potentials have been used to assess general brain and sensory function, and they are thought to reflect attention and memory processes engaged during stimulus processing [54,55]. In patients with decreased cognitive ability the P300 amplitude is smaller and the latency longer than in age-matched normal subjects [56,57]. Other authors, as results in this study, had reported a delay of P300 latency and smaller amplitudes in HIV-1 infected patients [58–69], that are detectable in the earliest/asymptomatic stages of HIV infection, getting worse as the infection progresses. Our results contribute additional information as co-infected patients presented longer visual P300 latencies at late stage and a smaller auditory P300 amplitude at early stage than HIV-1 infected/*T. gondii* non-infected patients, meaning that the simultaneous presence of HIV-1 and *T. gondii* create a poorer situation, further limiting cognitive processing capacity [70,71].

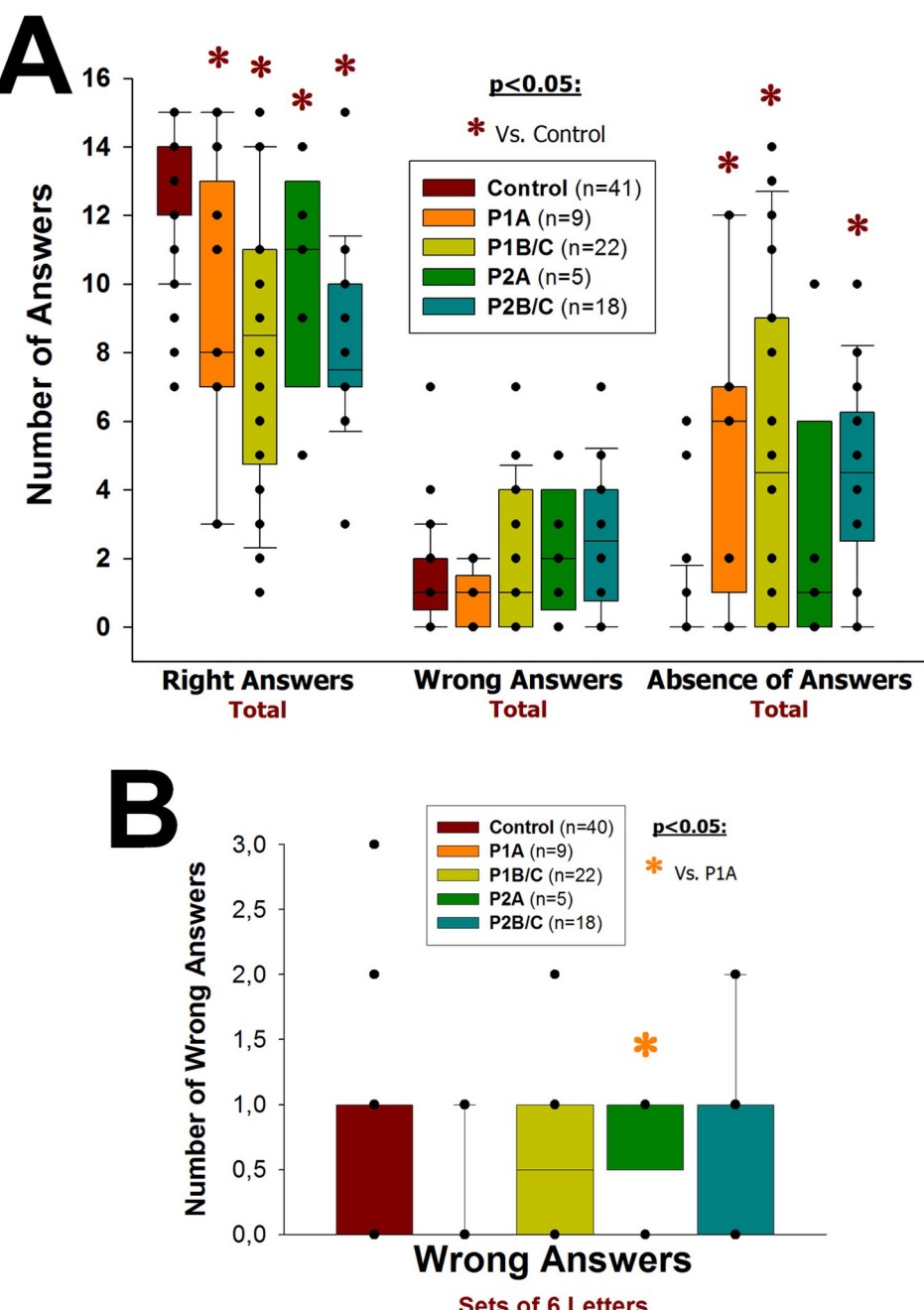

**Fig 5. Exploration of memory.** Vertical boxes with error bars represent medians, 25th and 75th percentiles of the number of right answers, wrong answers or absent answers in the evaluation of memory according to Sternberg [36], in HIV-1-infected (P1), HIV-1/*T. gondii* co-infected (P2) and HIV-1 non-infected (Control) individuals. Patients (P1 and P2) were grouped in early/asymptomatic or late/symptomatic, depending on whether their peripheral blood CD4 + T lymphocytes counts were higher (P1A and P2A) or lower (P1B/C and P2B/C) than 350/µL, respectively. Black dots represent every single patient, dark red asterisks (*) represent statistically significant differences between any given group of patients and the HIV-1 non-infected group in the same category of answers; orange asterisk (*) represents a statistically significant difference between P2A and P1A.

Regarding memory, several approaches have been described to explain its multiple components [36,72–77]. All groups of HIV-1-infected patients in this work performed significantly

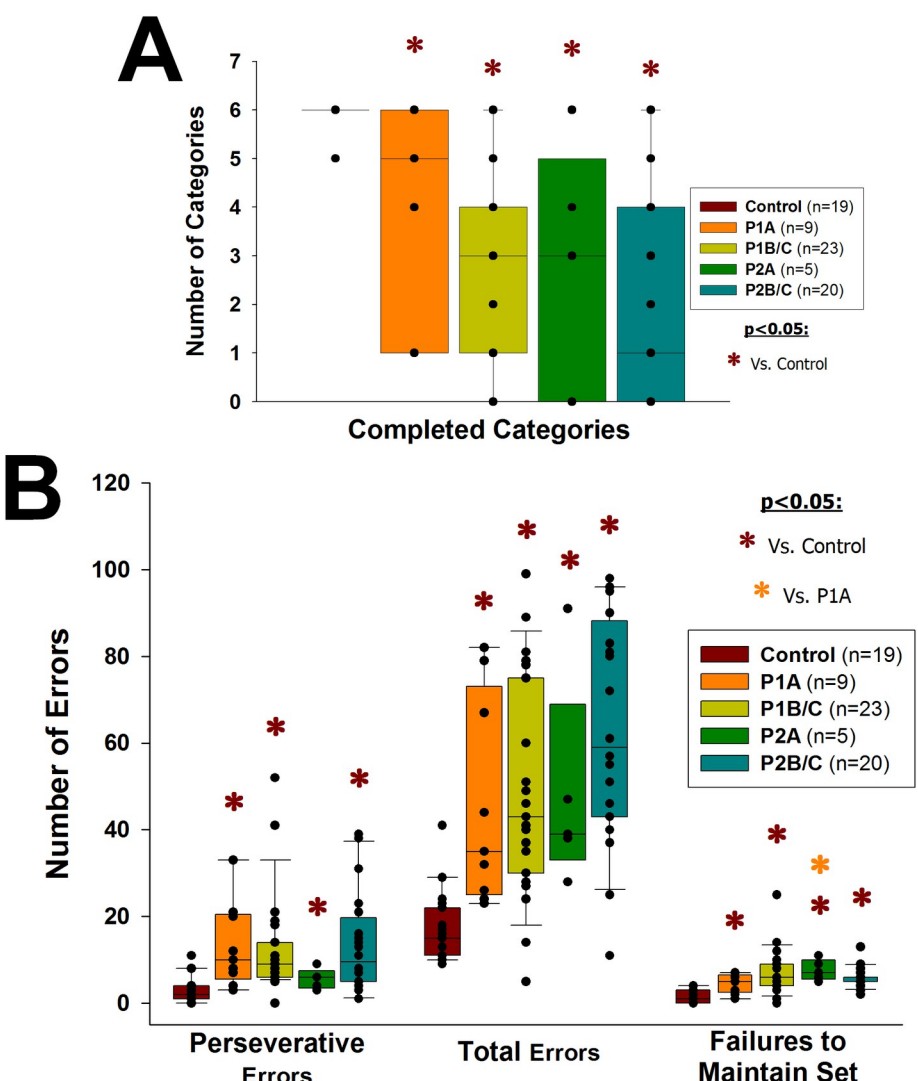

**Fig 6. Wisconsin Card Sorting Test.** Vertical boxes with error bars represent medians, 25th and 75th percentiles of the number of categories completed (A), perseverative errors, total errors and failures to maintain set (B) in the WCST, in HIV-1-infected (P1), HIV-1/*T. gondii* co-infected (P2) and HIV-1 non-infected (Control) individuals. Patients (P1 and P2) were grouped in early/asymptomatic or late/symptomatic, depending on whether their peripheral blood CD4 + T lymphocytes counts were higher (P1A and P2A) or lower (P1B/C and P2B/C) than 350/μL, respectively. Black dots represent every single patient, dark red asterisks (*) represent statistically significant differences between any given group of patients and the HIV-1 non-infected group; orange asterisk (*) represents a statistically significant difference between P2A and P1A.

below the seronegative group, in agreement with other studies [78–90], and again, it was noteworthy that co-infected patients in the early/asymptomatic stage of HIV-1 infection presented a significantly poorer performance in more complex tasks, suggesting additional limitations in initiating a systematic search of information in short-term/working memory [91]. Memory abilities are important for independent and successful management of activities of daily living [80,87], therefore, co-infected patients would be at higher risk of dependence on others in everyday functioning [87]. Several studies [79,82–84,86,90,92] have related memory impairment to prefrontostriatal circuit neuropathophysiology, secondary to HIV-1 infection. So, it is noteworthy that co-infected patients, who presented a poorer performance in more

complex tasks of Sternberg's memory evaluation, also presented smaller auditory P300 amplitude at prefrontal Fp1 location.

Executive functions are a set of general-purpose control processes that regulate the dynamics of human cognition (thoughts) and action (behaviors) [93], and therefore have broad implications for everyday life, and are related to self-control (willpower), health, wealth, and even public safety [94,95]. WCST was designed to assess problem solving, response maintenance and cognitive flexibility [37,96–98], and was used to evaluate these executive functions in the participants of this study. In agreement with previously reported works [99–103], all groups of HIV-1-infected patients presented a significantly poorer performance in WCST than HIV-1 non-infected group. Impaired WCST performance has been related to working memory deficits [104], as our results show. Moreover, co-infected patients also presented a significantly greater number of failure to maintain set, stressing the association of co-infection with early enhanced neurocognitive impairment.

Several mechanisms have been described [4,5,27,29,70,71,105–108] that could explain neurocognitive impairment described in this work. HIV-1 is able to invade the central nervous system early after systemic infection [109–111], productively infecting CNS-resident monocytic cells, which will release pro-inflammatory cytokines, viral proteins and chemokines, which in turn will activate uninfected macrophages and microglia to release potentially neurotoxic substances as quinolinic acid, excitatory amino acids, arachidonic acid, platelet-activating factor, free radicals and TNF-$\alpha$ [112]. These substances induce neuronal injury, synaptodendritic damage, and apoptosis [112]. HIV-1 infection of the brain also alters cell to cell communication systems, enhancing bystander apoptosis of uninfected cells, inflammation and viral infection [107]. All these mechanisms are responsible for neurodegeneration associated with HIV-1 infection that leads to neurocognitive impairment, even in the early stages. As HIV-1 infection progresses and secondary immunodeficiency (namely AIDS) appears, *T. gondii*-co-infected patients are in a higher risk of developing toxoplasmic encephalitis, with further CNS inflammation, necrosis of brain tissue and additional neurocognitive impairment [4,5,24–26,70,71] Moreover, this study contributes by reporting that co-infected patients have early defects in their immune response against *T. gondii*, and also show early (not just late) impaired P300 cognitive evoked potentials, memory and executive functions, with significantly poorer performance than that of patients infected only with HIV-1, and although a direct causal study was not done, evidence strongly suggests an association between early defects in immune response and enhancement of neurocognitive impairment in co-infected patients, even in the asymptomatic stage of HIV-1 infection. These results warrant reconsidering what "asymptomatic stage" really means. In fact, several groups are re-evaluating the classification of chronic *T. gondii* infection as asymptomatic, because cumulating evidence suggests that latent infection with the parasite is associated with a variety of neuropsychiatric and behavioural conditions [4,113–116]. Furthermore, results in this study emphasize the importance of early immune/neurocognitive evaluation, diagnosis and treatment of patients in order to ameliorate or even reverse potential impairment of neurocognitive function [80,117,118], allowing patients to successfully manage independent activities of daily living [87,100,102], and improving their quality of life.

## Supporting information

**S1 Table. Description of groups.**
(DOC)

**S2 Table. Production of cytokines.**
(DOC)

**S3 Table. Visual P300 latency.**
(DOC)

**S4 Table. Auditory P300 latency.**
(DOC)

**S5 Table. Auditory P300 amplitude.**
(DOC)

**S6 Table. Exploration of memory.**
(DOC)

**S7 Table. Wisconsin Card Sorting Test.**
(DOC)

**S8 Table. Production of cytokines–dispersion of values.**
(DOCX)

**S9 Table. Production of cytokines–p-values.**
(DOCX)

**S10 Table. Visual P300 latency–p-values.**
(DOCX)

**S11 Table. Auditory P300 latency–p-values.**
(DOCX)

**S12 Table. Auditory P300 amplitude–p-values.**
(DOCX)

**S13 Table. Exploration of memory–p-values.**
(DOCX)

**S14 Table. Wisconsin Card Sorting Test–p-values.**
(DOCX)

## Acknowledgments

Several members of our laboratories, including Wagner González, Edwin Díaz, Ydelys Fuentes, Riward Campelo, Alexandra Díaz, Josibel Camacho, Alexandra Rodríguez, Gustavo Rico, José Carrero, Wolfgang Vivas and Eduardo Navarro, participated evaluating patients and processing clinical samples.

## Author Contributions

**Conceptualization:** Edwin Eliel Escobar-Guevara, María Esther de Quesada-Martínez, Yhajaira Beatriz Roldán-Dávila, Belkisyolé Alarcón de Noya, Miguel Antonio Alfonzo-Díaz.

**Data curation:** Edwin Eliel Escobar-Guevara, María Esther de Quesada-Martínez.

**Formal analysis:** Edwin Eliel Escobar-Guevara, María Esther de Quesada-Martínez, Miguel Antonio Alfonzo-Díaz.

**Funding acquisition:** Edwin Eliel Escobar-Guevara, María Esther de Quesada-Martínez, Yhajaira Beatriz Roldán-Dávila, Belkisyolé Alarcón de Noya, Miguel Antonio Alfonzo-Díaz.

**Investigation:** Edwin Eliel Escobar-Guevara, María Esther de Quesada-Martínez, Miguel Antonio Alfonzo-Díaz.

**Methodology:** Edwin Eliel Escobar-Guevara, María Esther de Quesada-Martínez, Miguel Antonio Alfonzo-Díaz.

**Project administration:** Miguel Antonio Alfonzo-Díaz.

**Resources:** Yhajaira Beatriz Roldán-Dávila, Belkisyolé Alarcón de Noya, Miguel Antonio Alfonzo-Díaz.

**Supervision:** Miguel Antonio Alfonzo-Díaz.

**Validation:** María Esther de Quesada-Martínez.

**Writing – original draft:** Edwin Eliel Escobar-Guevara.

**Writing – review & editing:** Edwin Eliel Escobar-Guevara, María Esther de Quesada-Martínez, Yhajaira Beatriz Roldán-Dávila, Belkisyolé Alarcón de Noya, Miguel Antonio Alfonzo-Díaz.

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
