## [Decision Letter · Decision Letter 0]

23 May 2022

PONE-D-22-08134Defects in immune response to Toxoplasma gondii are associated with enhanced HIV-1-related neurocognitive impairment in co-infected patientsPLOS ONE

Dear Dr. Escobar,

Thank you for submitting your manuscript to PLOS ONE. After careful consideration, we feel that it has merit but does not fully meet PLOS ONE’s publication criteria as it currently stands. Therefore, we invite you to submit a revised version of the manuscript that addresses the points raised during the review process.

The data is exciting, however, the presentation is not well done and the interpretation is also compromised. Please answer all the issues. These article types are not expected to include results but may include pilot data. 

We look forward to receiving your revised manuscript.

Kind regards,

Eliseo A Eugenin, Ph.D.

Academic Editor

PLOS ONE

Journal Requirements:

Important: If there are ethical or legal restrictions to sharing your data publicly, please explain these restrictions in detail. Please see our guidelines for more information on what we consider unacceptable restrictions to publicly sharing data: http://journals.plos.org/plosone/s/data-availability#loc-unacceptable-data-access-restrictions. Note that it is not acceptable for the authors to be the sole named individuals responsible for ensuring data access

Reviewers' comments:

Reviewer's Responses to Questions

**Comments to the Author**

1. Is the manuscript technically sound, and do the data support the conclusions?

Reviewer #1: Partly

Reviewer #2: Yes

2. Has the statistical analysis been performed appropriately and rigorously? 

Reviewer #1: No

Reviewer #2: N/A

3. Have the authors made all data underlying the findings in their manuscript fully available?

Reviewer #1: Yes

Reviewer #2: Yes

4. Is the manuscript presented in an intelligible fashion and written in standard English?

Reviewer #1: No

Reviewer #2: Yes

5. Review Comments to the Author

Reviewer #1: The authors aim to investigate to neurocognitive impairment in individuals co-infected with HIV and Toxoplasma gondii. The study is straight forward and included the corresponding controls. However, there are problems associated to the organization and presentation of the data. The project would benefit of the following recommendations:

1. English language needs substantial revision.

2. Each graph should present every single subject on each bar graph. Also, the statitistical analyses need to be robust since this reviewer is skeptical that several comparisons are significant due to the high variability.

3. The manuscript needs standard submission organization. For example, each result subsection should state the main findings and each figure needs to include all the graphs for that specific figure (e.g., A, B, C,... F). Please see how other papers published in the journal are organized.

4. The discussion is really long and an iteration of the results. Therefore, please reduce it, eliminate figure labels in the discussion, summarize each result and compare it to other studies in the area.

Reviewer #2: In abstract session:

• For expression significantly, should be mention p-value.

• Statistical tests and software used for data analysis are mentioned.

In introduction session:

• The aim of the study should be clearly explain.

In methods session:

• Mention the study time.

• Mention the software used to analyze the data and its version.

• Mention the ethics code.

In results session:

• In some diagrams, did not mention the horizontal axis title.

• Abbreviations (C1, P1A, etc.) should be written in full below the image.

6. PLOS authors have the option to publish the peer review history of their article (what does this mean?). If published, this will include your full peer review and any attached files.

Reviewer #1: No

Reviewer #2: No

---

## [Author Response · Author response to Decision Letter 0]

12 Sep 2022

Response to Editor and Reviewers:

Answer: We have revised the entire document, trying to meet PLOS ONE's style requirements, including those for file naming.

2. We suggest you thoroughly copyedit your manuscript for language usage, spelling, and grammar. Upon resubmission, please provide the name of the colleague that edited your manuscript.

Answer: We were kindly assisted by a colleague in California, who prefers to remain anonymous, to read and proofread the entire manuscript, providing valuable suggestions for improvement.

3. In your Data Availability statement, you have not specified where the minimal data set underlying the results described in your manuscript can be found.

Answer: The data set used to reach the conclusions drawn in this manuscript is provided in S2-7 Tables (see Supporting Information). Two-tailed T-student or Mann-Whitney tests, as appropriate, were used to compare groups. Pearson product moment was used to analyze correlation. S8 Table provides medians, 25th and 75th percentiles for cytokines values. S9-14 Tables provides all particular p-values. Data in S2-7 Tables were used to build graphs.

4. Is the manuscript technically sound, and do the data support the conclusions?

a. Reviewer #1: Partly

Answer: We are providing additional Tables to explain how data were analyzed, and we hope that this additional information could show how data support the conclusions 

b. Reviewer #2: Yes

5. Has the statistical analysis been performed appropriately and rigorously?

a. Reviewer #1: No

Answer: We modified the presentation of results, showing p-values for statistically significant results, and provide S9-14 Tables with all particular p-values, including information about the method used and the power of performed test. We hope that this additional information could help to explain that statistical analysis has been performed appropriately and rigorously.

b. Reviewer #2: N/A

6. Have the authors made all data underlying the findings in their manuscript fully available?

a. Reviewer #1: Yes

b. Reviewer #2: Yes

7. Is the manuscript presented in an intelligible fashion and written in Standard English?

a. Reviewer #1: No

Answer: We have thoroughly copyedited our manuscript for language usage, spelling, and grammar. We were kindly assisted by a colleague in California, who read and edited the entire manuscript.

b. Reviewer #2: Yes

8. Review Comments to the Author

a. Reviewer #1: The authors aim to investigate to neurocognitive impairment in individuals co-infected with HIV and Toxoplasma gondii. The study is straight forward and included the corresponding controls. However, there are problems associated to the organization and presentation of the data. The project would benefit of the following recommendations:

i. English language needs substantial revision.

Answer: As we have mentioned before, we have thoroughly copyedited our manuscript for language usage, spelling, and grammar. A colleague in California read and copyedited the manuscript.

ii. Each graph should present every single subject on each bar graph. Also, the statistical analyses need to be robust since this reviewer is skeptical that several comparisons are significant due to the high variability.

Answer: We have modified graph presentation, including every single patient’s value as black dots. We provide every single p-value (S9-14

Tables), including information about the method used and the power of the performed test. We hope that this additional information could

explain that statistical analysis has been properly performed.

iii. The manuscript needs standard submission organization. For example, each result subsection should state the main findings and each figure needs to include all the graphs for that specific figure (e.g., A, B, C... F). Please see how other papers published in the journal are organized.

Answer: We have revised the entire document, trying to meet PLOS ONE's style requirements. We have modified the presentation of results, stating the main findings, showing p-values for statistically significant results and providing all particular p-values (S9-14 Tables). Figures presentation has been also modified.

iv. The discussion is really long and an iteration of the results. Therefore, please reduce it, eliminate figure labels in the discussion, summarize each result and compare it to other studies in the area.

Answer: We have revised and modified discussion, eliminating figure labels, summarizing results and comparing with other studies. We have

tried to reduce it, avoiding iteration, but trying to discuss all the important issues.

b. Reviewer #2:

i. In abstract session:

1. For expression significantly, should be mention p-value.

Answer: p<0.05 was considered as significant (see Abstract, page 3, line 48). We modified the presentation of results, showing p-values for

statistically significant results. All particular p-values are provided in S9-14 Tables.

2. Statistical tests and software used for data analysis are mentioned. 

Answer: Groups were compared using T-student or U-Mann-Whitney tests as appropriate (see Abstract, page 3, line 48). Software used to

perform statistical analysis of data was SigmaStat® for Windows, Version 1.0 (1992-1994 Jandel Corporation) (see Statistical analysis, page 13, lines 240, 241)

ii. In introduction session:

1. The aim of the study should be clearly explain.

Answer: The aim of the study was to evaluate how T. gondii-specific response changes in different stages of the HIV-1 infection, and if this

change could be associated with modifications in neurocognitive functions. Immune response was assessed by studying cytokine production in response to T. gondii antigens, and neurocognitive functions by evaluating auditory and visual P300 cognitive evoked potentials, Sternberg’s speed of memory scanning task and Wisconsin Card Sorting Test (see Introduction, pages 5 and 6, lines 91-94).

iii. In methods session:

1. Mention the study time.

Answer: Samples were analyzed and neurophysiological evaluations were performed between May 2007 and December 2009. Manuscript was written between August 2021 and February 2022.

2. Mention the software used to analyze the data and its version.

Answer: SigmaStat® for Windows, Version 1.0 (1992-1994 Jandel Corporation) was used to perform statistical analysis of data

3. Mention the ethics code.

Answer: All participants provided written informed consent and the study was approved by the ethic boards of the participating institutions, in accordance with the Declaration of Helsinki (1964-2013) and the Belmont Report (1979).

iv. In results session:

1. In some diagrams, did not mention the horizontal axis title. 

Answer: Figures presentation has been revised and modified. 

2. Abbreviations (C1, P1A, etc.) should be written in full below the image.

Answer: Figures legends explain what abbreviations mean: HIV-1-non infected individuals that were seronegative (C1) or seropositive (C2) to T. gondii, and HIV-1 infected (P1) or HIV-1/T. gondii co-infected (P2) patients. Patients (P1 and P2) were grouped as early/asymptomatic or late/symptomatic, depending if their peripheral blood CD4+ T lymphocytes counts were higher (P1A and P2A) or lower (P1B/C and P2B/C) than 350/μL, respectively.

---

## [Decision Letter · Decision Letter 1]

28 Nov 2022

PONE-D-22-08134R1Defects in immune response to Toxoplasma gondii are associated with enhanced HIV-1-related neurocognitive impairment in co-infected patientsPLOS ONE

Dear Dr. Escobar,

Thank you for submitting your manuscript to PLOS ONE. After careful consideration, we feel that it has merit but does not fully meet PLOS ONE’s publication criteria as it currently stands. Therefore, we invite you to submit a revised version of the manuscript that addresses the points raised during the review process.

Dear Dr. Escobar 

Thank you for submit your manuscript to PLOsone. The reviewers and I, feel that you did not include all the answers and changed the figures as requested. You and your team needs to include all the changes suggested

Best Regards

Eliseo Eugenin

We look forward to receiving your revised manuscript.

Kind regards,

Eliseo A Eugenin, Ph.D.

Academic Editor

PLOS ONE

Dear Dr. Escobar

Thank you for submit your manuscript to PLOsone. The reviewers and I, feel that you did not include all the answers and changed the figures as requested. You and your team needs to include all the changes suggested

Best Regards

Eliseo Eugenin

Reviewers' comments:

Reviewer's Responses to Questions

**Comments to the Author**

1. If the authors have adequately addressed most of the comments raised in a previous round of review 

Reviewer #1: (No Response)

Reviewer #2: 

2. Is the manuscript technically sound, and do the data support the conclusions?

Reviewer #1: Partly

Reviewer #2: Yes

3. Has the statistical analysis been performed appropriately and rigorously? 

Reviewer #1: Yes

Reviewer #2: I Don't Know

4. Have the authors made all data underlying the findings in their manuscript fully available?

Reviewer #1: Yes

Reviewer #2: Yes

5. Is the manuscript presented in an intelligible fashion and written in standard English?

Reviewer #1: No

Reviewer #2: Yes

6. Review Comments to the Author

Reviewer #1: The authors should check the format of other PLoS One published papers and use similar format. Also, the results' presentation is boring. The authors should use the conclusions of each section as title of each result subsection instead of "Production of IL-10". Figure legends should be in their own subsection at the end of the manuscript in order instead of being inserted between the result section. The discussion for the results obtanied should not be longer than 4 pages long. Presently, the discussion is 8 pages long which is ridiculous. Please be succinct and avoid iteration of the results.

Reviewer #2:  They need to include the suggested changes 

7. PLOS authors have the option to publish the peer review history of their article (what does this mean?). If published, this will include your full peer review and any attached files.

Reviewer #1: No

Reviewer #2: No

---

## [Author Response · Author response to Decision Letter 1]

28 Feb 2023

Response to Editor and Reviewers:

Comments to the Author

1. If the authors have adequately addressed most of the comments raised in a previous round of review

Reviewer #1: (No Response)

Reviewer #2:

2. Is the manuscript technically sound, and do the data support the conclusions?

Reviewer #1: Partly

Answer: We have revised the entire document, and made deep modifications in results and discussion sections. All data used to reach the conclusions drawn in this manuscript is provided. Tables are also provided to explain how data were analyzed, to show how data support conclusions.

Reviewer #2: Yes

3. Has the statistical analysis been performed appropriately and rigorously?

Reviewer #1: Yes

Reviewer #2: I Don't Know

Answer: We show p-values for statistically significant results, and provide S9-14 Tables with all p-values, including information about the method used and the power of performed test. This information explain that statistical analysis been performed appropriately and rigorously.

4. Have the authors made all data underlying the findings in their manuscript fully available?

Reviewer #1: Yes

Reviewer #2: Yes

5. Is the manuscript presented in an intelligible fashion and written in standard English?

Reviewer #1: No

Answer: We kindly asked Laura Bartoli, MD, from Australia and other colleague from California, who prefers to remain anonymous, to read and proofread the entire manuscript. They provided suggestions for improvement.

Reviewer #2: Yes

6. Review Comments to the Author

Reviewer #1:

• The authors should check the format of other PLoS One published papers and use similar format.

o Answer: We checked other PLoS One published papers to use a similar format in our manuscript.

• Also, the results' presentation is boring.

o Answer: we modified results section, to help reader to obtain the information easily.

• The authors should use the conclusions of each section as title of each result subsection instead of "Production of IL-10". 

o Answer: We modified results section, stating the main findings of each subsection.

• Figure legends should be in their own subsection at the end of the manuscript in order instead of being inserted between the result section. 

o Answer: we had followed the instructions given to authors in the Submission Guidelines document:

“Figure captions

Figure captions must be inserted in the text of the manuscript, immediately following the paragraph in which the figure is first cited”. (https://journals.plos.org/plosone/s/submission-guidelines#loc-figures-and-tables

• The discussion for the results obtained should not be longer than 4 pages long. Presently, the discussion is 8 pages long which is ridiculous. Please be succinct and avoid iteration of the results.

o Answer: we substantially reduced the length of discussion section to meet, as close as possible, the required four pages for discussion.

Reviewer #2:

• They need to include the suggested changes

o Answer: we had made our best effort to include all suggested changes.

---

## [Decision Letter · Decision Letter 2]

7 May 2023

Defects in immune response to Toxoplasma gondii are associated with enhanced HIV-1-related neurocognitive impairment in co-infected patients

PONE-D-22-08134R2

Dear Dr. Escobar-Guevara,

We’re pleased to inform you that your manuscript has been judged scientifically suitable for publication and will be formally accepted for publication once it meets all outstanding technical requirements.

Kind regards,

Eliseo A Eugenin, Ph.D.

Academic Editor

PLOS ONE

Additional Editor Comments (optional):

Dear Dr. Escobar-Guevara

Thank you for submit your corrections to PLOSone. My apologies for the extended time for an answer.

Eliseo Eugenin

Reviewers' comments:

Reviewer's Responses to Questions

**Comments to the Author**

1. If the authors have adequately addressed your comments raised in a previous round of review and you feel that this manuscript is now acceptable for publication, you may indicate that here to bypass the “Comments to the Author” section, enter your conflict of interest statement in the “Confidential to Editor” section, and submit your "Accept" recommendation.

Reviewer #1: All comments have been addressed

2. Is the manuscript technically sound, and do the data support the conclusions?

Reviewer #1: Yes

3. Has the statistical analysis been performed appropriately and rigorously? 

Reviewer #1: Yes

4. Have the authors made all data underlying the findings in their manuscript fully available?

Reviewer #1: Yes

5. Is the manuscript presented in an intelligible fashion and written in standard English?

Reviewer #1: Yes

6. Review Comments to the Author

Reviewer #1: (No Response)

7. PLOS authors have the option to publish the peer review history of their article (what does this mean?). If published, this will include your full peer review and any attached files.

Reviewer #1: No

---

## [Editor Report · Acceptance letter]

15 May 2023

PONE-D-22-08134R2 

Defects in immune response to *Toxoplasma gondii* are associated with enhanced HIV-1-related neurocognitive impairment in co-infected patients 

Dear Dr. Escobar-Guevara:

I'm pleased to inform you that your manuscript has been deemed suitable for publication in PLOS ONE. Congratulations! Your manuscript is now with our production department. 

Kind regards, 

on behalf of

Dr. Eliseo A Eugenin 

Academic Editor

PLOS ONE